# Palladium Decorated, Amine Functionalized Ni-, Cd- and Co-Ferrite Nanospheres as Novel and Effective Catalysts for 2,4-Dinitrotoluene Hydrogenation

**DOI:** 10.3390/ijms232113197

**Published:** 2022-10-30

**Authors:** Viktória Hajdu, Emőke Sikora, Ferenc Kristály, Gábor Muránszky, Béla Fiser, Béla Viskolcz, Miklós Nagy, László Vanyorek

**Affiliations:** 1Institute of Chemistry, University of Miskolc, Miskolc-Egyetemváros, 3515 Miskolc, Hungary; 2Institute of Mineralogy and Geology, University of Miskolc, Miskolc-Egyetemváros, 3515 Miskolc, Hungary

**Keywords:** ferrite, magnetic catalyst, hydrogenation, TDA, nanostructure

## Abstract

2,4-diaminotoluene (TDA) is one of the most important polyurethane precursors produced in large quantities by the hydrogenation of 2,4-dinitrotoluene using catalysts. Any improvement during the catalysis reaction is therefore of significant importance. Separation of the catalysts by filtration is cumbersome and causes catalyst loss. To solve this problem, we have developed magnetizable, amine functionalized ferrite supported palladium catalysts. Cobalt ferrite (CoFe_2_O_4_-NH_2_), nickel ferrite (NiFe_2_O_4_-NH_2_), and cadmium ferrite (CdFe_2_O_4_-NH_2_) magnetic catalyst supports were produced by a simple coprecipitation/sonochemical method. The nanospheres formed contain only magnetic (spinel) phases and show catalytic activity even without noble metals (palladium, platinum, rhodium, etc.) during the hydrogenation of 2,4-dinitrotoluene, 63% (*n*/*n*) conversion is also possible. By decorating the supports with palladium, almost 100% TDA selectivity and yield were ensured by using Pd/CoFe_2_O_4_-NH_2_ and Pd/NiFe_2_O_4_-NH_2_ catalysts. These catalysts possess highly favorable properties for industrial applications, such as easy separation from the reaction medium without loss by means of a magnetic field, enhanced reusability, and good dispersibility in aqueous medium. Contrary to non-functionalized supports, no significant leaching of precious metals could be detected even after four cycles.

## 1. Introduction

The catalytic hydrogenation of 2,4-dinitrotoluene (2,4-DNT) is one of the most widely used industrial processes for the synthesis of 2,4-toluenediamine (2,4-TDA). TDA is an important intermediate in the formation of toluene diisocyanate (TDI, 2.49 Mtons produced in 2021) used in the production of polyurethanes, a compound used primarily in flexible polyurethane foams, elastomers, coatings, and adhesives. Transition metals (Pd, Pt, Ni, etc.) or transition metal oxides on carbon support (carbon nanotubes, graphene, carbon black) are most commonly used as catalysts in the catalytic hydrogenation of DNT [1,2,3,4,5,6,7]. Their advantage is that they adsorb the various organic substances well and bind the catalytically active metal particles (platinum, palladium). Furthermore, they are well dispersed in a liquid medium, which makes it easy for the reactant molecules to have access to the catalytically active metals on the surface of the support particles. Specific surface area is a key factor in heterogeneous catalysis, which increases dramatically with decreasing particle size according to the square-cubic law. However, catalysts of small particle size form a stable dispersion in the reaction medium, therefore their economical recovery cannot be achieved without loss by conventional filtration and centrifugation operations. This issue can be almost completely eliminated by using magnetizable catalysts, efficient catalyst recovery can be achieved by a simple magnetic separation operation [8]. Possible supports with magnetic properties are: ferrites, chromium dioxide, magnetite, maghemite. They are also used in many separation operations, such as DNA purification [9,10], heavy metal ions binding [11,12], removal of organic contaminants from water [13,14], and even in the field of catalysis [15,16,17,18]. Weng and co-workers [19] converted nickel-laden electroplating slurry into NiFe_2_O_4_ nanomaterial using sodium carbonate by a hydrothermal washing strategy. The prepared nanoparticles showed stable electrochemical Li storage performance. This new strategy can provide a sustainable approach for the conversion of heavy metals in industrial waste into high-value functional materials and for the selective recycling of heavy metals. Ebrahimi and co-workers [20] prepared superparamagnetic CoFe_2_O_4_NPs@Mn-Organic Framework core-shell nanocomposites by a layer-by-layer method. The structures exhibit high temperature stability and good magnetization. This magnetic nanometal-organic framework is an excellent candidate in targeted drug-delivery systems.

It is important to ensure that the metal oxide with magnetic properties as a catalyst support is well dispersed in the reaction medium. Since magnetic nanoparticles have a strong agglomeration tendency, in order to solve this problem, it is necessary to modify the surface of the magnetic nanoparticles with different functional groups (NH_2_, OH, SiH, SH groups) [21,22,23,24,25,26]. In some cases, without surface modification, the products of hydrogenolysis can deactivate the supported precious metal catalysts due to their poisoning effects, therefore a higher amount of catalyst is necessary to complete the reaction [27,28]. Since heterogeneous catalysis involves the adsorption of reactants (intermediates and products as well), surface functionality may be crucial for reaching good conversion and selectivity. Amino-groups on the surface may promote the binding of nitro-compounds (reactant), via H-bonding interactions. In addition, the functional groups can substantially enhance the binding of catalytically active noble metals on the surface during the preparation of the catalyst [29]. Sharma and co-workers have successfully prepared a supported ruthenium nanoparticle on amino-functionalized Fe_3_O_4_ (Fe_3_O_4_/NH_2_/Ru) that exhibits excellent catalytic activity for the transfer hydrogenation of nitro compounds using NaBH_4_ as hydrogen donor. In addition, the catalyst can be easily recovered by an external magnetic field, recycled five times and reused without loss of activity [30].

In the preparation of conventional (non-magnetic) oxide- (SiO_2_, Al_2_O_3_ etc.) or C-based catalysts, after the catalytically active metal salt or its oxide has been deposited on the support surface, heat treatment in an inert atmosphere (nitrogen, argon) is required, followed by an activation step (most often reduction with hydrogen gas) to form an elemental catalytically active form of the precious metal. As this last step takes place at high temperatures (300–400 °C), this process is not feasible in the case of metal oxides with magnetic properties (magnetite, maghemite, chromium dioxide, ferrites) because they are not thermally stable at such high temperatures. In this case, an alternative catalyst preparation method should be used. Precious metal deposition by sonochemistry on the surface of ferrite particles may be an ideal solution. Thus, ultrasonic cavitation in the alcoholic solution of noble metal precursors reduces the noble metal nanoparticles on the surface of the ferrite particles [31]. Intense ultrasonic irradiation yields vapor bubbles, or rather cavities in the liquid as the boiling point decreases. These vapor bubbles collapse in the fluid in the high pressure ranges, releasing large amounts of energy in these microvolumes, which can cover the energy requirements of chemical reactions [32]. This way, catalytically active magnetic nanoparticles containing noble metals can be efficiently produced.

The aim of this work Is the development of stable and selective catalysts of high activity, which are easily separable from the reaction-medium, without loss. Hereby, we report the preparation and possible application of amine-functionalized magnetic, Co-, Ni-, or Cd-ferrite based catalysts decorated with Pd. The composition, morphology, and surface of the nanoparticles have been examined in detail. In addition, their applicability in the industrially important hydrogenation of 2,4-dinitrotoluene has been studied.

## 2. Results and Discussion

The amine-functionalized cobalt-, cadmium-, and nickel ferrite nanoparticles were investigated by HRTEM (Figure 1a–c). The HRTEM pictures clearly show the spherical shape of the ferrite nanoparticles, which are composed of smaller, individual nanoparticles of size 4–10 nm (Appendix A). The average crystallite size of the ferrite nanoparticles (the building blocks of the spherical aggregates) was also calculated based on the XRD patterns by the mean column length calibrated method using the full width at half maximum (FWHM) and the width of the Lorentzian component of the fitted profiles. The sizes of the individual nanoparticles, which can be measured in the HRTEM pictures are coherent with the XRD measurements. The average size of the nanoparticles, which build up the NiFe_2_O_4_-NH_2_ is 6 ± 2 nm. In the case of the CoFe_2_O_4_-NH_2_ and CdFe_2_O_4_-NH_2_ nanospheres (building blocks), the average sizes are 4 ± 2 nm and 8 ± 3 nm, respectively.

In contrast, the average particle sizes of the aggregate ferrite spheres, namely CoFe_2_O_4_-NH_2_, NiFe_2_O_4_-NH_2_, and CdFe_2_O_4_-NH_2_ are 51 ± 8 nm, 42 ± 13 nm, and 80 ± 14 nm (Figure 1d and Table 1). The particle size distribution is broad in all three cases, based on the interquartile range width in the box plot diagram in Figure 1d. The cadmium-ferrite sample contains the biggest aggregates; however, the mean and median particle sizes are very close to each other, these are 80 ± 14 nm and 84 nm (Table 1).

In heterogeneous catalysis the specific surface area of the nanoparticles (supports) is a key feature, since it affects the sorption of the different chemical species during the reaction. The specific surface areas of the ferrites were determined by carbon dioxide adsorption measurements using the Dubinin–Astakhov model. The highest surface area was measured in case of the CoFe_2_O_4_-NH_2_ (279 m^2^/g), much higher than those of the other two samples: 72 m^2^/g (CdFe_2_O_4_-NH_2_) and 93 m^2^/g (NiFe_2_O_4_-NH_2_) The specific surface areas are below that of the activated carbon supports (up to 1000 m^2^/g).

XRD measurements revealed the spinel structure of the prepared magnetic particles (Figure 1e). The following reflexions can be identified on the diffractograms: (111), (220), (311), (400), (422), (511), and (440) at 18.1°, 30.1°, 35.5°, 43.2°, 53.8°, 57.2°, and 62.8° two Theta degrees which belong to NiFe_2_O_4_ spinel (PDF 10-0325). In the case of the CoFe_2_O_4_ phase the above listed reflexions are located at 18.3°, 30.3°, 35.4°, 43.1°, 53.6°, 57.0°, and 62.5° two Theta degrees (PDF 22-1086). The reflexions of the CdFe_2_O_4_ phase are found at 18.2°, 29.8°, 35.0°, 42.6°, 53.0°, 56.4°, and 61.9° two Theta degrees (PDF 89-2810). Based on the XRD analysis we can say that the synthetized ferrite samples contained only the appropriate spinel phase, other non-magnetic oxides are not found in addition to the ferrites.

Electron diffraction measurements on the individual ferrite nanospheres further supported the exclusive presence of spinel phases, since the measured d spacing were perfectly correlated with the d-values in X-ray databases (Figure 2).

To map the surface functionality and structure of the amine functionalized ferrite nanoparticles Fourier-transform infrared (FTIR) spectra were recorded. In the spectra presented in Figure 3a the following bands can be identified: two bands between 500 cm^−1^ and 600 cm^−1^ and between 400 cm^−1^ and 450 cm^−1^ which belong to the tetrahedral and octahedral complexes of the spinel structures, respectively [33,34]. The band at 592 cm^−1^ belongs to the vibration of Fe^3+^–O^2−^ in the sublattice A-site. The presence of the absorption band at 416 cm^−1^ can be assigned to the trivalent metal-oxygen vibration at the octahedral B-sites [33,34]. The band at 1052 cm^−1^ belongs to C-N stretching in the case of the three ferrites [35,36]. In the case of all ferrite samples, a band at 1615 cm^−1^ can be identified as the N–H stretching vibration of the free amino functional groups. Additional absorption bands are found at 871 cm^−1^, 1048 cm^−1^, 2874 cm^−1^, and 2929 cm^−1^ which belong to the C–O vibrations, the C–N, and symmetric and asymmetric C–H stretching vibrations, respectively [35,36]. The presence of νC-O and νC-H bands suggest that adsorbed organic molecules (ethylene glycol and ethanol amine) were anchored on the surface of the ferrite particles. The stretching vibration band of the N–H bonds overlaps with the vibration band of -OH groups. The bending vibration mode of the -OH groups resulted a band at 1393 cm^−1^. As a result of the polar functional groups on the surface, the magnetic ferrite nanoparticles are easily dispersed in aqueous medium. Moreover, they can be easily removed upon the action of an external magnetic field (Figure 3b). Additional advantage of the amine functional groups is that they improve the stabilization and distribution of the Pd nanoparticles, simultaneously as electron donor, the amino groups may enhance the surface electron density on the palladium particles.

Bonds containing carbon can be identified in the FTIR spectra, thus the total carbon content of the palladium-decorated ferrite samples was examined by CHNS elemental analysis. The carbon content was the highest in the case of the Pd/NiFe_2_O_4_-NH_2_ (6.3 wt%), the other two catalysts contain lower amounts of carbon: 2.5 wt% (Pd/CoFe_2_O_4_-NH_2_) and 2.9 wt% (Pd/CdFe_2_O_4_-NH_2_) (Table 2). The presence of carbon can be explained by the anchoring of the ethylene glycol and ethanol amine molecules on the surface of the ferrite nanoparticles. The total nitrogen content, which originates from the amino functional groups, was also measured. The highest N content was found in the case of the nickel-ferrite supported catalyst: 1.4 wt%, while 0.4 wt% and 0.5 wt% was found in the case of the cobalt and cadmium ferrites, respectively (Table 2). The palladium content of the freshly prepared catalysts was measured by ICP-OES. The Pd contents were found to be 5.05 wt% (Pd/CoFe_2_O_4_-NH_2_), 5.25 wt% (Pd/CdFe_2_O_4_-NH_2_) and 3.96 wt% (Pd/NiFe_2_O_4_-NH_2_).

The palladium containing ferrite catalysts were investigated by HRTEM (Figure 4a–c). In the TEM pictures, the same spherical aggregates as the original catalyst-support can be identified. It should be noted here that the aggregates did not disintegrate despite the high-energy sonication during the palladium anchoring on ferrites. The phase identification of the Pd/NiFe_2_O_4_-NH_2_, Pd/CoFe_2_O_4_-NH_2_, and Cd/NiFe_2_O_4_-NH_2_ catalysts was carried out based on XRD measurements (Figure 4d–f). In addition to the nickel-ferrite, cobalt-ferrite, and the cadmium-ferrite magnetic catalyst-supports elemental palladium could also be identified on the diffractograms. The (111) and (200) reflexions of the elemental palladium are located at 40.2° and 46.7° two Theta degrees in the case of all catalysts (PDF 46-1043). The ultrasound-assisted decomposition of the palladium, did not change the chemical composition of the ferrite phases, indicating that the ferrites have sufficient chemical stability.

The visual identification of palladium nanoparticles is difficult and uncertain next to the ferrite nanoparticles due to their small particle size, thus HAADF pictures and element mapping were made to confirm their presence (Figure 5 and Appendix A). In the HAADF pictures, the Pd nanoparticles are slightly brighter compared to the ferrite nanoparticles. Moreover, the element mapping also confirmed the position of the palladium particles next to the ferrites. Palladium nanoparticles (highlighted in yellow) are located on the surface of the spherical CoFe_2_O_4_ aggregates (Figure 5) as was also confirmed in the case of Pd/CdFe_2_O_4_-NH_2_ and Pd/NiFe_2_O_4_-NH_2_ (Appendix A). The presence of cobalt and iron are also detectable and are marked with green and red color.

The particle size of the Pd nanoparticles was measured in the HAADF pictures using ImageJ software (version 1.53t) and the scalebars (Appendix A). The measured diameters of Pd particles were 5.1 ± 0.6 nm (Pd/CoFe_2_O_4_-NH_2_), 3.8 ± 0.5 nm (Pd/CdFe_2_O_4_-NH_2_), and 4.0 ± 0.8 nm (Pd/NiFe_2_O_4_-NH_2_). Very similar particle sizes were resulted based on the XRD measurements, 4 ± 2 nm, 6 ± 2 nm, and 4 ± 2 nm in case of the cobalt-ferrite, cadmium-ferrite, and nickel-ferrite-supported palladium catalysts.

### Catalytic Tests of the Magnetic Catalysts for Hydrogenation of 2,4-DNT

Before the catalytic measurements of the palladium decorated, amine-functionalized ferrites, the palladium-free magnetic supports were tested in the hydrogenation of 2,4-DNT at 333 K temperature for 240 min. The tests revealed that even without the noble metal the amine-functionalized cobalt ferrite, cadmium ferrite, and nickel ferrite catalyst supports showed activity in TDA synthesis. The highest DNT conversion (63.01% *n*/*n*) was achieved in the case of the CoFe_2_O_4_-NH_2_ sample. The other two spinels were less active, 19.4% *n*/*n* (CdFe_2_O_4_-NH_2_) and 26.7% *n*/*n* (NiFe_2_O_4_-NH_2_) DNT conversions were measured. In order to increase the DNT conversion and TDA yield, the use of palladium is necessary.

The catalytic activity of the palladium decorated ferrite supported catalysts was compared in the hydrogenation of 2,4-dinitrotoluene to 2,4-diaminotoluene at three different reaction temperatures (at 303 K, 313 K, and 333 K). Interestingly, total DNT conversion was achieved in a short time (40 min) by using the Pd/CoFe_2_O_4_-NH_2_ and Pd/NiFe_2_O_4_-NH_2_ catalysts at 333 K hydrogenation temperature (Figure 6a,c). In addition, close to or above 99 *n*/*n*% TDA yields were reached after 120 min at 333 K in the case of the Pd/CoFe_2_O_4_-NH_2_ (98.9%) and Pd/NiFe_2_O_4_-NH_2_ (99.9%) catalysts (Figure 6d,f). The catalytic activity of the Pd/CdFe_2_O_4_-NH_2_ sample was found to be far below the two above mentioned catalysts (Figure 6b,e). More than 99 *n*/*n*% DNT conversion was reached after four hours, and the maximum TDA yield was only 65.5 *n*/*n*% after 240 min at 333 K.

GC-MS measurements revealed the presence of two semi-hydrogenated intermediates, namely 4-amino-2-nitrotoluene (4A2NT) and 2-amino-4-nitrotoluene (2A4NT), which were not transformed to 2,4-TDA in the case of the Pd/CdFe_2_O_4_ catalyst. By using the cobalt- or nickel-based catalysts (Pd/CoFe_2_O_4_-NH_2_ and Pd/NiFe_2_O_4_-NH_2_) the two semi-hydrogenated compounds were completely transformed to 2,4-TDA after 120 min hydrogenation at 333 K, resulting a >99 *n*/*n*% selectivity (2,4-TDA) for both catalysts (Figure 7). In contrast, the selectivity was only 65% for the cadmium ferrite-supported catalyst. Byproducts were not detectable after the tests.

Since the Pd/CoFe_2_O_4_-NH_2_ and Pd/NiFe_2_O_4_-NH_2_ catalysts had the highest DNT conversion, TDA yield, and selectivity, these two catalysts were selected for reuse-tests. The catalysts were tested in four cycles at 333 K temperature at 20 bar hydrogen pressure. As can be seen in Figure 8, the results are almost identical for each reuse cycle (4), i.e., there is no visible difference during the whole reaction time between the DNT conversions and TDA yields. The catalytic activity remained constant and excellent even after repeated use. The results suggest that this type of catalyst is stable, namely no Pd loss occurred, possibly due to a strong interaction between the ferrite catalyst support and the palladium particles. The results of ICP-OES measurements are in line with those of the reuse experiments, since the palladium content of the reused catalysts was almost the same, as that of the fresh (non-used) Pd/NiFe_2_O_4_-NH_2_ and Pd/CoFe_2_O_4_-NH_2_ samples (Table 3). This catalyst stability and the no apparent loss of the noble metal may be explained by the presence of the amine functional groups anchoring the Pd nanoparticles on the ferrite nanospheres. This theory is further supported by our previous results, where significant loss in catalytic activity and noble metal was observed in the case of non-functionalized ferrites [36].

The activity, yield, and selectivity of our catalysts are as high as of those found in the literature. Malyala and co-workers [37] investigated a powdered Y zeolite (10% Ni/HY) catalyst containing 10% Ni for the hydrogenation of 2,4 DNT with 85% TDA yield and selectivity. Ren et al. [38] investigated several Pd/C and Pt/ZrO_2_ catalysts. The Pd/C catalyst’s yield and selectivity were 98% and 99.2%, respectively. The Pt/ZrO_2_ catalysts yield varied between 97.1% and 98.9% depending upon Pt-content and reduction temperature. The selectivity values are the same as yields since conversion was 100% in each case. In addition, our catalysts are magnetic, therefore they can be easily separated from the reaction medium and even the support itself shows catalytic activity.

## 3. Materials and Methods

The amine-functionalized ferrite nanoparticles were made from nickel(II) nitrate hexahydrate, Ni(NO_3_)_2_∙6H_2_O, MW: 290.79 g/mol (Thermo Fisher GmbH, D-76870 Kandel, Germany), cadmium(II) nitrate tetrahydrate, Cd(NO_3_)_2_∙4H_2_O, MW: 308.46 g/mol (Acros Organics Ltd., B-2440 Geel, Belgium), cobalt(II) nitrate hexahydrate, Co(NO_3_)_2_∙6H_2_O, MW: 291.03 g/mol and iron(III) nitrate nonahydrate, Fe(NO_3_)_3_∙9H_2_O, MW: 404.00 g/mol (VWR Int. Ltd., B-3001 Leuven, Belgium), respectively. As dispersion medium ethylene glycol, HOCH_2_CH_2_OH, (VWR Int. Ltd., F-94126 Fontenay-sous-Bois, France) was applied. For coprecipitation and functionalization of the ferrites, ethanolamine, NH_2_CH_2_CH_2_OH (Merck KGaA, D-64271 Darmstadt, Germany) and sodium acetate, CH_3_COONa (ThermoFisher GmbH, D-76870 Kandel, Germany) were used. Palladium(II) nitrate dihydrate, Pd(NO_3_)_2_∙2H_2_O, MW: 266.46 g/mol (Thermo Fisher Scientific Ltd., D-76870 Kandel, Germany) as Pd precursor and Patosolv^®^, a mixture of 90 vol% ethanol and 10 vol% isopropanol (Molar Chem. Ltd., H-2314 Halásztelek, Hungary) were used during the preparation of the magnetic catalysts. The starting material, main product, and the standards used for the catalytic tests and calibration of the GC measurements were as follows: 2,4-dinitrotoluene (DNT), C_7_H_6_N_2_O_4_, MW:182.13 g/mol, 2,4-diaminotoluene (TDA), C_7_H_10_N_2_, MW: 122.17 g/mol, 4-methyl-3-nitroaniline, 2-methyl-3-nitroaniline and 2-methyl-5-nitroaniline, C_7_H_8_N_2_O_2_, MW: 152.15 g/mol, (Sigma Aldrich Chemie Gmbh, D-89555 Steinheim, Germany). As an internal standard, nitrobenzene, C_6_H_5_NO_2_, MW: 123.11 g/mol (Merck KGaA, D-64293 Darmstadt, Germany) was used.

### 3.1. Preparation of the Amine-Functionalized Magnetic Spinel Nanoparticles and the Pd-Catalyst

Amine-functionalized ferrite magnetic catalyst supports were synthesized according to a modified coprecipitation method. In 50 mL ethylene glycol iron(III) nitrate nonahydrate and the nitrate salt of the respective transition metal (Co, Ni or Cd) were dissolved (Table 3). Sodium acetate 12.30 g (15 mmol) was dissolved in another 100 mL ethylene-glycol and it was heated to 100 °C in three necked flask under reflux and continuous stirring. The solution of the metal precursors was added to the glycol-based sodium acetate solution, followed by the addition of 35 mL ethanol amine. After 12 h continuous agitation and reflux, the cooled solution was separated by centrifugation (4200 rpm, 10 min). The solid phase was washed by distilled water several times and the magnetic ferrite was easily separated by a magnet from the aqueous phase. Finally, the ferrite was also rinsed with anhydrous ethanol, and was dried by lyophilization. These ferrite samples were used as magnetic catalyst support for the preparation of palladium decorated spinel catalysts.

For the deposition of the palladium nanoparticles onto the surface of the ferrite crystals, a Hielscher UIP100 Hdt homogenizer was used (120 W, 17 kHz). First, palladium(II) nitrate (0.25 g) was dissolved in patosolv (50 mL) containing 2.00 g dispersed ferrite. Then the solution was treated by ultrasonic cavitation for two minutes. During the process, elemental palladium nanoparticles formed from the Pd(II)-ions as a result of the reducing action of the alcohol (patosolv).

### 3.2. Catalytic Test: Hydrogenation of 2,4-Dintitro Toluene

2,4-dinitrotoluene, DNT (c_n_:50 mmol/dm^3^ in methanolic solution, V_tot_ = 150 mL) was hydrogenated by using 0.10 g magnetic Pd catalysts in a Büchi Uster Picoclave reactor of 200 mL volume under constant agitation at 1000 rpm (Appendix A).

The pressure of the hydrogenation was kept at 20 bar in all experiments and the reaction temperature was set to 303 K, 313 K, and 333 K, respectively. The sampling was carried out after 0, 5, 10, 15, 20, 30, 40, 60, 80, 120, 180, and 240-min hydrogenation. As internal standard, 5.0 µL nitrobenzene was added to 1.00 mL sample.

### 3.3. Characterization Techniques

High-resolution transmission electron microscopy (HRTEM, Talos F200X G2 electron microscope with field emission electron gun, X-FEG, accelerating voltage: 20–200 kV) was applied for examination of particle size and morphology. For the imaging and electron diffraction a SmartCam digital search camera (Ceta 16 Mpixel, 4 k × 4 k CMOS camera) and high-angle annular dark-field (HAADF) detector were used. For the TEM measurement, the samples were dispersed in distilled water, and this aqueous dispersion was dropped on 300 mesh copper grids (Ted Pella Inc., 4595 Redding, CA 96003, USA). The qualitative and quantitative analysis of the ferrite and the palladium catalysts were carried out by X-ray diffraction (XRD) measurements by applying the Rietveld method. Bruker D8 diffractometer (Cu-Kα source) in parallel beam geometry (Göbel mirror) with Vantec detector was used. The average crystallite size of the oxide domains was calculated from the mean column length calibrated method by using full width at half maximum (FWHM) and the width of the Lorentzian component of the fitted profiles. The carbon content originating from the residual ethanolamine and ethylene glycol on the surface of the ferrites, was determined by a Vario Macro CHNS element analyzer by using the phenanthrene standard (C: 93.538%, H: 5.629%, N: 0.179%, S: 0.453%) from Carlo Erba Inc. Helium (99.9990%) was the carrier gas, and oxygen (99.995%) was used as the oxidative atmosphere. The functional groups on the surface of the amine-functionalized ferrite nanoparticles were identified by Fourier transform infrared spectroscopy (FTIR), with a Bruker Vertex 70 spectroscope. Spectra were recorded in transmission mode, in KBr pellet (10 mg ferrite sample was pelletized with 250 mg potassium bromide). The palladium contents of the magnetic ferrite catalysts were measured by a Varian 720 ES inductively coupled optical emission spectrometer (ICP-OES). For the ICP-OES measurements, the catalysts were dissolved in aqua regia. The specific surface area of the catalysts was examined based on the Dubinin–Astakhov (DA) method, by CO_2_ adsorption–desorption experiments at 273 K using a Micromeritics ASAP 2020 sorptometer. The 2,4-toluenediamine containing samples after the hydrogenation tests were analyzed using an Agilent 7890A gas chromatograph coupled with Agilent 5975C Mass Selective detector. RTX-1MS column (30 m × 0.25 mm × 0.25 μm) was applied and the injected sample volume was 1 μL at 200:1 split ratio, while the inlet temperature was set to 473 K. Helium was used as a carrier gas (2.28 mL/min), and the oven temperature was set to 323 K for 3 min and it was heated up to 523 K with a heating rate of 10 K/min and kept there for another 3 min.

The catalytic activity of the magnetic Pd catalysts was compared based on calculating the conversion (*X*%) of 2,4-dinitrotoluene as follows (Equation (1)):(1)$$X\%=\frac{\;n_{DNT\left(consumed\right)\;}}{n_{DNT\left(initial\right)\;}}\;\cdot \;100\;$$

The yield (*Y*%) of TDA was calculated based on the following equation (Equation (2)):(2)$$Y\%=\frac{n_{TDA\left(formed\right)}}{n_{TDA\left(theoretical\right)\;}}\;\cdot \;100$$
where ***n****_TDA_* is the corresponding molar amount of the product (2,4-diaminotoluene).

Aniline selectivity (*S*%) of the catalysts was also calculated by using the DNT conversion and TDA yield (Equation (3)):(3)$$S\%=\frac{Y}{X}\cdot 100$$

## 4. Conclusions

Amine-functionalized, cobalt-, nickel-, and cadmium ferrite magnetic catalyst supports were synthesized based on a modified coprecipitation method. HRTEM investigations showed that the support nanoparticles are spherical and composed of smaller individual nanoparticles of size 4–10 nm. In contrast, the average particle sizes of the aggregate ferrite spheres, namely CoFe_2_O_4_-NH_2_, NiFe_2_O_4_-NH_2_, and CdFe_2_O_4_-NH_2_ are 50.9 ± 8.2 nm, 41.5 ± 12.9 nm, and 80.2 ± 14.2 nm. Based on XRD and electron diffraction measurements, the exclusive presence of spinel phases could be detected without any non- magnetic oxides. The surface of the nanoparticles contains NH_2_-groups, which promote the dispersibility of the particles in aqueous medium. The good dispersibility is a key factor in catalytic applications, since aggregation reduces the active surface of the particles. Nevertheless, the magnetic nanoparticles are easily recollected from the reaction medium by the action of an external magnetic field, which can greatly reduce operating costs and catalyst loss during separation. The palladium-free magnetic supports were tested in the hydrogenation of 2,4-DNT and moderate catalytic activity in TDA synthesis was found. The highest DNT conversion (63.01% *n*/*n*) was achieved in the case of the CoFe_2_O_4_-NH_2_ sample, indicating that the use of palladium is necessary. Palladium nanoparticles were deposited onto the ferrite surfaces utilizing a fast and facile sonochemical method, yielding an immediately usable, catalytically active form. The presence of palladium dramatically increased the catalytic performance. Total DNT conversion was achieved in a short time (40 min) by using the Pd/CoFe_2_O_4_-NH_2_ and Pd/NiFe_2_O_4_-NH_2_ catalysts at 333 K hydrogenation temperature, while TDA yields were (98.9%) and (99.9%) for the cobalt- and nickel-based catalysts after 120 min at 333 K.

No drop in performance was observed during reuse tests, as the conversion, yield, and selectivity values remained unchanged and excellent for four catalytic cycles. ICP-OES measurements revealed no loss in the palladium content of the reused catalysts. This catalyst stability and the no apparent loss of the noble metal may be explained by the presence of the amine functional groups anchoring the Pd nanoparticles on the ferrite nanospheres. In summary, amine functionalization of magnetic ferrite supports can significantly enhance catalytic performance and reusability. Based on our results, the above detailed magnetically separable catalysts may be well used for the hydrogenation of other aromatic nitro compounds.

## Figures and Tables

**Figure 1 ijms-23-13197-f001:**
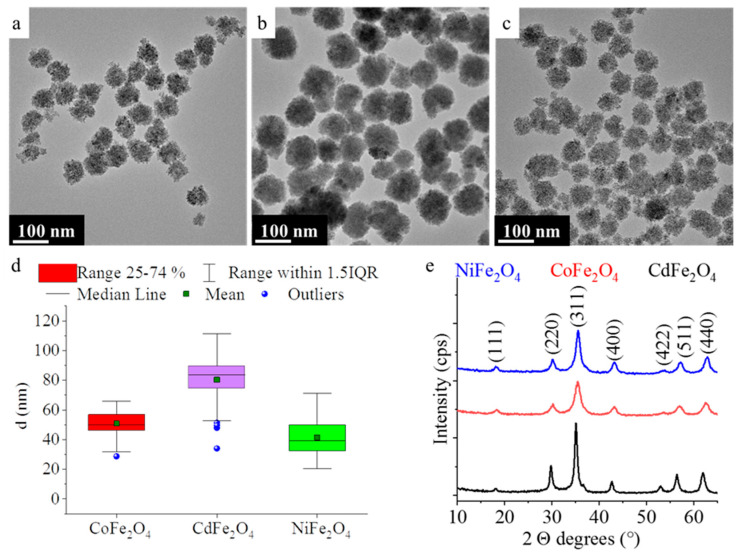
HRTEM pictures of the CoFe_2_O_4_-NH_2_ (**a**), CdFe_2_O_4_-NH_2_ (**b**) and NiFe_2_O_4_-NH_2_ (**c**). Box plot diagrams for the particle size analysis (**d**) and the XRD pattern of the ferrites (**e**).

**Figure 2 ijms-23-13197-f002:**
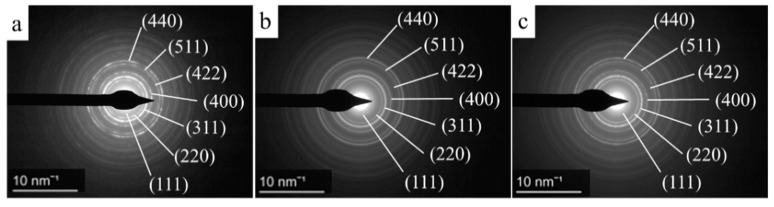
Electron diffraction patterns of the CoFe_2_O_4_-NH_2_ (**a**), CdFe_2_O_4_-NH_2_ (**b**) and NiFe_2_O_4_-NH_2_ (**c**) with the Miller indexes (PDF 89-2810; PDF 22-1086 and PDF 10-0325).

**Figure 3 ijms-23-13197-f003:**
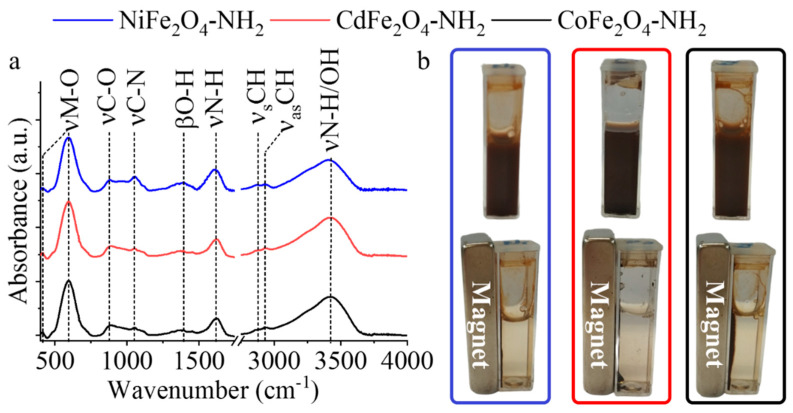
(**a**) FTIR spectra of the NiFe_2_O_4_-NH_2_, CdFe_2_O_4_-NH_2_ and CoFe_2_O_4_-NH_2_ supports. (**b**) Demonstration of the good dispersibility and separability of the nanoparticles by magnetic field.

**Figure 4 ijms-23-13197-f004:**
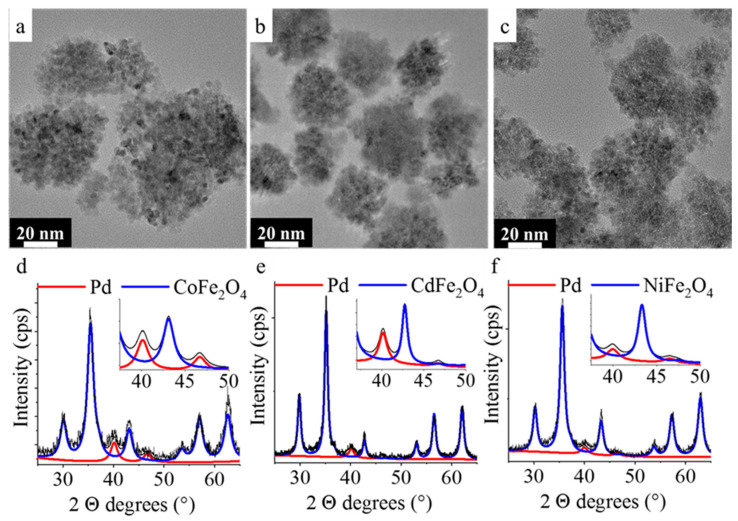
HRTEM pictures and XRD patterns of the Pd/CoFe_2_O_4_-NH_2_ (**a**,**d**), Pd/CdFe_2_O_4_-NH_2_ (**b**,**e**), and Pd/NiFe_2_O_4_-NH_2_ catalysts (**c**,**f**).

**Figure 5 ijms-23-13197-f005:**
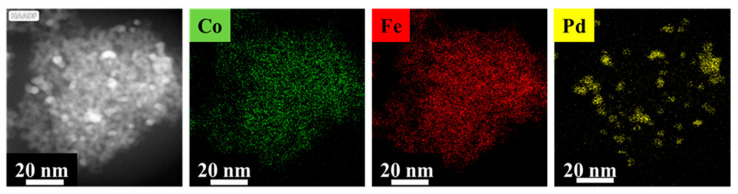
Element maps of the Pd/CoFe_2_O_4_-NH_2_ catalyst.

**Figure 6 ijms-23-13197-f006:**
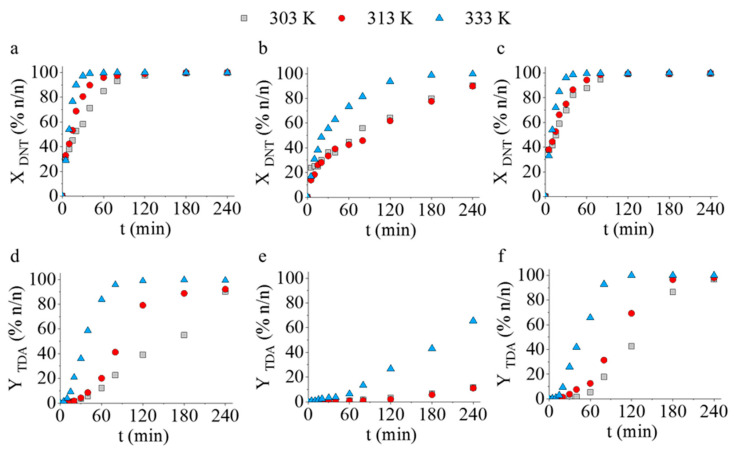
DNT conversions and TDA yields vs. time of hydrogenation at 303 K, 313 K, and 333 K using Pd/CoFe_2_O_4_-NH_2_ (**a**,**d**), Pd/CdFe_2_O_4_-NH_2_ (**b**,**e**), and Pd/NiFe_2_O_4_-NH_2_ (**c**,**f**) catalysts.

**Figure 7 ijms-23-13197-f007:**
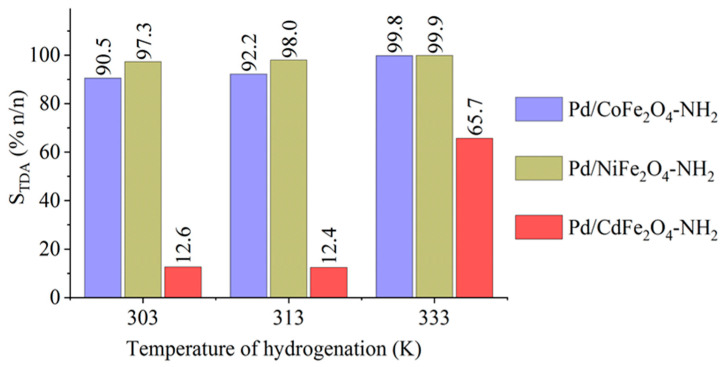
The maximum selectivity of TDA at three reaction temperature for the magnetic catalysts.

**Figure 8 ijms-23-13197-f008:**
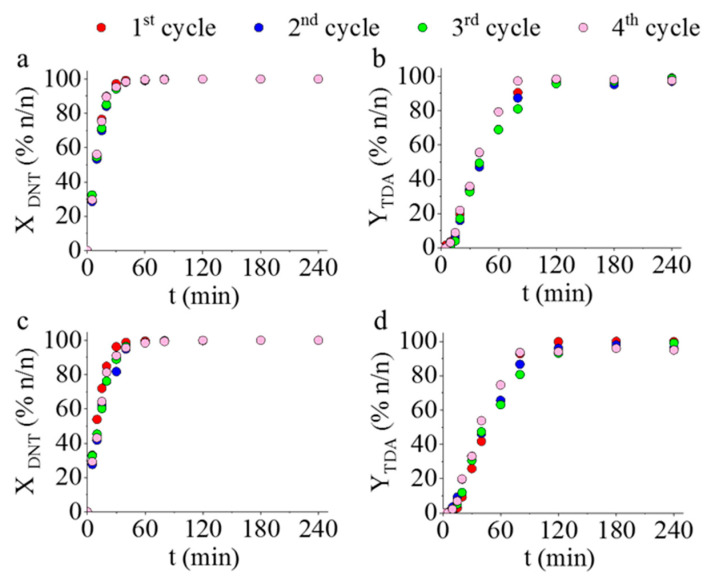
Reuse tests of the Pd/CoFe_2_O_4_-NH_2_ (**a**,**b**) and Pd/NiFe_2_O_4_-NH_2_ (**c**,**d**) catalysts. Change of the 2,4-DNT conversion and 2,4-TDA yield vs. time of hydrogenation.

**Table 1 ijms-23-13197-t001:** Results of the size analysis (in nm) of the ferrite nanospheres (based on HRTEM pictures).

(nm)	Mean	SD	Min.	Max.	1st Quartile	3rd Quartile	Median	P90	P95
CoFe_2_O_4_-NH_2_	50.9	8.2	28.8	66.0	46.3	56.9	50.0	61.2	62.6
CdFe_2_O_4_-NH_2_	80.2	14.2	34.0	111.4	74.5	89.5	83.7	92.1	100.7
NiFe_2_O_4_-NH_2_	41.5	12.9	20.5	71.2	32.4	50.0	39.3	61.6	66.4

**Table 2 ijms-23-13197-t002:** Carbon and nitrogen content (based on CHNS elemental analysis) of the catalysts, as well as their palladium content before use and after four catalytic cycles.

	C wt%	N wt%	Pd wt% (Before Use)	Pd wt% (After Use)
Pd/CoFe_2_O_4_-NH_2_	2.5	0.4	5.05	4.68
Pd/CdFe_2_O_4_-NH_2_	2.9	0.5	5.25	5.10
Pd/NiFe_2_O_4_-NH_2_	6.3	1.4	3.96	3.78

**Table 3 ijms-23-13197-t003:** Amounts of the respective transition metal nitrates dissolved in the reaction media for the synthesis of the NH_2_-functionalized ferrite nanoparticles.

	Ni(NO_3_)_2_∙6H_2_O	Cd(NO_3_)_2_∙4H_2_O	Co(NO_3_)_2_∙6H_2_O	Fe(NO_3_)_3_∙9H_2_O
NiFe_2_O_4_-NH_2_	2.91 g (1 mmol)	-	-	8.08 g (2 mmol)
CdFe_2_O_4_-NH_2_	-	3.08 g (1 mmol)	-	8.08 g (2 mmol)
CoFe_2_O_4_-NH_2_	-	-	2.91 g (1 mmol)	8.08 g (2 mmol)

## Data Availability

Data is available upon request from the corresponding authors.

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
