# Peer review of "Palladium Decorated, Amine Functionalized Ni-, Cd- and Co-Ferrite Nanospheres as Novel and Effective Catalysts for 2,4-Dinitrotoluene Hydrogenation"

_ijms, 2022, doi:10.3390/ijms232113197_

Round 1
Reviewer 1 Report
The research topic is good and author describe it well.
Author Response
Thank You for Your positive comments! We appreciate it!
Reviewer 2 Report
The work is well written and could be published after minor revision.
1) Page 2, line 97 - check the formula of ethanolamine
2) Typically, values and confidence intervals are rounded to significant numbers. For example, 80.2 ± 14.2 nm should be rounded up to 80 ± 14 nm - there will be no significant difference. I also advise to round to integers the values of the specific surface area (as well as selectivity of TDA)
3) In some cases, the names of catalysts and carriers contain amino groups, and in some cases they do not. If these are the same materials, then make the names the same
4) "Palladium nanoparticles are not visually distinguishable from ferrite nanoparticles 273 due their small particle size, thus element mapping was carried out for the confirmation 274 of their presence"- well, I can quite easily distinguish palladium nanoparticles from ferrite particles.Palladium particles are much darker in TEM images
5) What is the reason for the significantly lower activity of Pd/CdFe2O4-NH2 compared to other catalysts?
6) You write about amino-modified spinels - and how exactly is the surface of spinels modified? Are amino-groups covalently bonded with spinels? Or is there absorption/adsorption of ethanolamine? Also, amino groups can be attached to the surface of the spinels and not interact with palladium. Discuss these questions at work.
7)In this work, the size of spinel crystallites was estimated from XRD reflections. Can the same be done with palladium nanoparticles? Are the results consistent with the TEM data?
Author Response
Reviewer #2
Comments and Suggestions for Authors
The work is well written and could be published after minor revision.
Thank You for Your positive comments! We appreciate it!
Point 1: Page 2, line 97 – check the formula of ethanolamine
Thank You for the remark! The formula has been corrected.
Point 2: Typically, values and confidence intervals are rounded to significant numbers. For example, 80.2 ± 14.2 nm should be rounded up to 80 ± 14 nm – there will be no significant difference. I also advise to round to integers the values of the specific surface area (as well as selectivity of TDA)
The values and confidence intervals have been rounded to integers.
Point 3: In some cases, the names of catalysts and carriers contain amino groups, and in some cases they do not. If these are the same materials, then make the names the same
Thank you for pointing this out! The names were corrected (unified) in the text.
Point 4: "Palladium nanoparticles are not visually distinguishable from ferrite nanoparticles 273 due their small particle size, thus element mapping was carried out for the confirmation 274 of their presence"- well, I can quite easily distinguish palladium nanoparticles from ferrite particles.Palladium particles are much darker in TEM images
Response 4:
The above sentence was modified as follows:
„The visual identification of palladium nanoparticles is difficult and uncertain next to the ferrite nanoparticles due to their small particle size, thus HAADF pictures and element mapping were made to confirm their presence (Fig.5, FigS6 a and b, in the SI). In the HAADF pictures, the Pd nanoparticles are slightly brighter compared to the ferrite nanoparticles. Moreover, the element mapping also confirmed the position of the palladium particles next to the ferrites.”
Point 5: What is the reason for the significantly lower activity of Pd/CdFe2O4-NH2 compared to other catalysts?
Response 5:
The lower activity may be explained by the unique electron configuration (d10s2) of cadmium among transition metals. Like mercury and zinc, cadmium has a closed d subshell, while cobalt and nickel have empty (partially filled) d orbitals. d-pi interactions are essential for a good catalytic activity. Since cobalt and the nickel have a partially filled d sub-shell in the metallic state, these can form d-pi bonds with electron rich molecules, in our case with the DNT. In the case of cadmium, the fully filled d orbitals cannot overlap with the pi orbitals reducing its catalytic activity. In summary, the different electron configuration may explain the different catalytic activity.
Point 6: You write about amino-modified spinels - and how exactly is the surface of spinels modified? Are amino-groups covalently bonded with spinels? Or is there absorption/adsorption of ethanolamine? Also, amino groups can be attached to the surface of the spinels and not interact with palladium. Discuss these questions at work.
Response 6:
On the FTIR spectra of the ferrites, we identified stretching vibration bands of the C-O and C-H bonds, which suggests the presence of adsorbed organic compounds on surface of the ferrite particles (ethylene glycol and monoethanol amine). CHNS element analysis revealed ~ 2-6 wt% carbon content (Table 3), which is in line with the presence of adsorbed organic molecules. Based on this, it can be stated, that for anchoring the amino groups on the ferrite, adsorbed ethanolamine is also needed.
The following lines were inserted into the text:
“The presence of νC-O and νC-H bands suggest that adsorbed organic molecules (ethylene glycol and ethanol amine) were anchored on the surface of the ferrite particles.”
„Additional advantage of the amine functional groups is that they improve the stabilization and distribution of the Pd nanoparticles, simultaneously as electron donor, the amino groups may enhance the surface electron density on the palladium particles.”
Point 7: In this work, the size of spinel crystallites was estimated from XRD reflections. Can the same be done with palladium nanoparticles? Are the results consistent with the TEM data?
Response 7:
Yes, of course. The Pd size were also calculated based on the XRD reflection. Moreover, in the HAADF TEM pictures of the catalysts, the palladium particles were measured by ImageJ program using the scalebar. These results were added in the manuscript and also in the Supplementary Information.
„The particle size of the Pd nanoparticles was measured in the HAADF pictures using ImageJ software and the scalebars (FigS7, FigS8 and FigS9). The measured diameters of Pd particles were 5.1 ± 0.6 nm (Pd/CoFe2O4-NH2), 3.8 ± 0.5 nm (Pd/CdFe2O4-NH2) and 4.0 ± 0.8 nm (Pd/NiFe2O4-NH2). Very similar particle sizes were obtained based on the XRD measurements, 4 ± 2 nm, 6 ± 2 nm and 4 ± 2 nm in the case of the cobalt-ferrite, cadmium-ferrite and nickel-ferrite supported palladium catalysts.”
Reviewer 3 Report
In this manuscript, the authors have presented simple preparation of magnetic catalysts based on cobalt, nickel, or cadmium ferrites modified with palladium nanoparticles. The catalytic activity of the entitles catalysts was tested in the reaction of hydrogenation of 2,4-dinitrotoluene with the formation of 2,4-dinitrotoluene – an important compound used as precursor for preparation of toluene diisocyanate. The simplicity of the approach is an indisputable advantage of this work, and high efficiency complements the overall picture. I believe, that the study is well designed and the presented data are solid and meet all the requirements of the journal. I recommend the publication of this work in International Journal of Molecular Sciences after minor revisions and some clarifications.
1) In-text consecutive references must be combined in groups, e.g., [1-8].
2) page 2, lines 59-60. Why do you need to bind amino compounds (products) after the reaction?
3) Please, carefully check all sub/superscripted indexes.
4) page 2, lines 68-71. What do you mean by "conventional oxide"?
5) page 3, line 116. "The solution of the metal ions" sounds odd, please, rephrase.
6) Check the temperatures used in catalytic tests –213 K, 323 K, and 333 K in experimental part and 303 K, 313 K and 333 K in results and discussion. Why did you use 323 K when studied palladium-free magnetic supports??
7) Any idea, why cobalt-ferrite has so high surface area in comparison with other ferrites?
8) The band at 1393 cm-1 was attributed to βO-H, while in text it is attributed to C-N.
9) page 7, line 244. What do you mean by "Bonds containing carbon can be identified in the FTIR spectra"?
10) Do you have any explanations why Pd/NiFe2O4, containing the highest amount of carbon and nitrogen, has the lowest amount of palladium?
11) There is no any comparison of the efficiency of the material obtained with the literature data.
12) Please check the English, and especially the punctuation.

Author Response
Response to Reviewer 3 Comments
Comments and Suggestions for Authors
In this manuscript, the authors have presented simple preparation of magnetic catalysts based on cobalt, nickel, or cadmium ferrites modified with palladium nanoparticles. The catalytic activity of the entitles catalysts was tested in the reaction of hydrogenation of 2,4-dinitrotoluene with the formation of 2,4-dinitrotoluene – an important compound used as precursor for preparation of toluene diisocyanate. The simplicity of the approach is an indisputable advantage of this work, and high efficiency complements the overall picture. I believe, that the study is well designed and the presented data are solid and meet all the requirements of the journal. I recommend the publication of this work in International Journal of Molecular Sciences after minor revisions and some clarifications.
Thank You for your valuable comments. We believe they will help to improve the quality of our manuscript!
Point 1: In-text consecutive references must be combined in groups, e.g., [1-8].
Response 1: Thank you the suggestion, the references has been corrected.
Point 2: page 2, lines 59-60. Why do you need to bind amino compounds (products) after the reaction?
Response 2:
Thank you for the remark, we clarified our statement. The advantage of the hydrogen bond between the nitro-compounds and the amino functional groups of the catalyst is that it improves the adsorption of the reactant molecules. The text has been changed as follows:
„Amino-groups on the surface may promote the binding of nitro-compounds (reactant), via H-bonding interactions.”
Point 3: Please, carefully check all sub/superscripted indexes.
Response 3: Thank you for the remark, we corrected all sub/superscripted indexes.
Point 4: page 2, lines 68-71. What do you mean by "conventional oxide"?
Response 4
By conventional oxides, we mean the non-magnetic oxide catalyst supports that are still used today, such as aluminum oxide, silicon dioxide. We clarified this in the text.
Point 5: page 3, line 116. "The solution of the metal ions" sounds odd, please, rephrase.
Response: 5
It was rephrased. „The solution of the metal precursors was added into the glycol-based sodium acetate solution, followed by the addition of 35 ml ethanol amine.”
Point 6: Check the temperatures used in catalytic tests –213 K, 323 K, and 333 K in experimental part and 303 K, 313 K and 333 K in results and discussion. Why did you use 323 K when studied palladium-free magnetic supports??
Response 6
The correct temperatures are 303 K, 313 K and 333 K. These were corrected in both the experimental section and Catalytic test sections.
Point 7: Any idea, why cobalt-ferrite has so high surface area in comparison with other ferrites?
Response 7
Cobalt ferrite was synthesized under the same reaction conditions as the other two ferrites. We do not know the reason for the said discrepancy.
Point 8: The band at 1393 cm-1 was attributed to βO-H, while in text it is attributed to C-N.
Response 8:
Thank you for the remark!
The vawenumber 1393 cm-1 belongs to the bending vibration of the hydroxyl groups. The postion of νC-N band is at 1048 cm-1. These were corrected in the manuscript as:
“Additional absorption bands are found at 871 cm-1, 1048 cm-1, 2874 cm-1 and 2929 cm−1…”
“The bending vibration mode of the -OH groups resulted a band at 1393 cm-1.”
Point 9: page 7, line 244. What do you mean by "Bonds containing carbon can be identified in the FTIR spectra"?
Response 9:
On the FTIR spectra of the ferrites, we identified stretching vibration bands of the C-O and C-H bonds, which suggests the presence of adsorbed organic compounds on surface of the ferrite particles (ethylene glycol and monoethanol amine). CHNS element analysis revealed ~ 2-6 wt% carbon content (Table 3), which is in line with the presence of adsorbed organic molecules.
The following lines were inserted into the text:
“The presence of νC-O and νC-H bands suggest that adsorbed organic molecules (ethylene glycol and ethanol amine) were anchored on the surface of the ferrite particles.”
Point 10: Do you have any explanations why Pd/NiFe2O4, containing the highest amount of carbon and nitrogen, has the lowest amount of palladium?
Response 10
Adsorption of palladium ions can be different on a carbon surface than on a ferrite surface, this can cause a difference in the palladium content. Moreover, in the case of palladium particles, crystal nucleation and crystal growth on oxide surfaces are favored. Due to the higher carbon content, the oxide surface was less accessible to palladium ions, so it could bind to the carbon as well.
Point 11: There is no any comparison of the efficiency of the material obtained with the literature data.
Response 11
In addition to the very high TDA selectivity and yield, the stability of the amine functionalized Pd/NiFe2O4 and Pd/CoFe2O4 catalysts is also remarkable. Compared to the amine functionalized ferrite supported catalyst, other ferrite supported catalyst (Pd/ZnFe2O4), show significant palladium leaching after the reuse tests. In the case of the non-functionalized Pd/ZnFe2O4 catalyst palladium loss after the 4th cycle was observed, the initial palladium content decreased from 4.20 wt% to 1.8 wt% [https://doi.org/10.1016/j.jmrt.2022.06.113]. This palladium loss led to a decrease in the catalytic activity during the reuse tests.
Moreover, the following section was added in the manuscript right before the Conclusions section for comparison of the experimental results.
“The activity, yield and selectivity of our catalysts are as high as of those found in the literature. Malyala and co-workers [37] investigated a powdered Y zeolite (10 % Ni/HY) catalyst containing 10 % Ni for the hydrogenation of 2,4 DNT with 85 % TDA yield and selectivity. Ren et al [38] investigated several Pd/C and Pt/ZrO2 catalysts. The Pd/C catalyst’s yield and selectivity were 98% and 99.2%, respectively. The Pt/ZrO2 cat-alysts yield varied between 97.1% and 98.9% depending upon Pt-content and reduction temperature. The selectivity values are the same as yields since conversion was 100% in each case. In addition, our catalysts are magnetic, therefore they can be easily separated from the reaction medium and even the support itself shows catalytic activity.”
Point 12: Please check the English, and especially the punctuation.
Response 12: Done.
Reviewer 4 Report
The manuscript presents a new approach for the hydrogenation of 2,4-dinitro- 3 toluene using palladium decorated, amine-functionalized Ni-, Cd- and Co-ferrite nanoparticles. The presentation and characterization are informative and convincing. Overall, the results are good and I would like to recommend publishing after minor modifications.
The Pd dispersion is not great as shown in “Figure 5. Element maps of the Pd/CoFe2O4 catalyst.” and “FigS4: Element mapping of the Pd/CdFe2O4-NH2 (a) and Pd/NiFe2O4-NH2 (b)”. The authors should discuss how the dispersion would affect the catalytic results. In addition, the Pd% is too high which deteriorates the significance of these catalysts; I cannot see the effect of other metals (the selectivity difference from 97.3 to 99.9 may already be within the experimental error range). The authors used both yields and selectivity in the figures and discussion. Please be coherent throughout the work.
Author Response
Response to Reviewer 4 Comments
Comments and Suggestions for Authors
The manuscript presents a new approach for the hydrogenation of 2,4-dinitro- 3 toluene using palladium decorated, amine-functionalized Ni-, Cd- and Co-ferrite nanoparticles. The presentation and characterization are informative and convincing. Overall, the results are good and I would like to recommend publishing after minor modifications.
Thank You for your valuable comments. We believe they will help to improve the quality of our manuscript!
Point 1: The Pd dispersion is not great as shown in “Figure 5. Element maps of the Pd/CoFe2O4 catalyst.” and “FigS4: Element mapping of the Pd/CdFe2O4-NH2 (a) and Pd/NiFe2O4-NH2 (b)”.
Response 1:
The Pd size were also calculated based on the XRD reflection. Moreover, in the HAADF TEM pictures of the catalysts, the palladium particles were measured by ImageJ program using the scalebar. These results were added in the manuscript and also in the Supplementary Information.
„The particle size of the Pd nanoparticles was measured in the HAADF pictures using ImageJ software and the scalebars (FigS7, FigS8 and FigS9). The measured diameters of Pd particles were 5.1 ± 0.6 nm (Pd/CoFe2O4-NH2), 3.8 ± 0.5 nm (Pd/CdFe2O4-NH2) and 4.0 ± 0.8 nm (Pd/NiFe2O4-NH2). Very similar particle sizes were obtained based on the XRD measurements, 4 ± 2 nm, 6 ± 2 nm and 4 ± 2 nm in the case of the cobalt-ferrite, cadmium-ferrite and nickel-ferrite supported palladium catalysts.”
Point 2: The authors should discuss how the dispersion would affect the catalytic results.
Response 2: The importance of good dispersibility is highlighted in the Introduction, starting from line 62:
“It is important to ensure that the metal oxide with magnetic properties as a catalyst support is well dispersed in the reaction medium. Since magnetic nanoparticles have a strong agglomeration tendency, in order to solve this problem, it is necessary to modify the surface of the magnetic nanoparticles with different functional groups (NH2, OH, SiH, SH groups)[21–26].”
And also in line 382: “The good dispersibility is a key factor in catalytic applications, since aggregation reduces the active surface of the particles.”
Point 3: In addition, the Pd% is too high which deteriorates the significance of these catalysts; I cannot see the effect of other metals (the selectivity difference from 97.3 to 99.9 may already be within the experimental error range).
Response 3: The reviewer is right, from an economic point of view the palladium content should be optimized and the precious metal should be used as little as possible. This will be part of a later study. The 5 wt% palladium content was chosen, because the commercially available catalysts also have a palladium content of 5% (or more). Indeed, there is no great difference between the efficiency of nickel-ferrite and cobalt-ferrite supported catalysts in terms of conversions, so only these catalysts were considered efficient and investigated during the reuse tests.
The catalytic effect of the catalyst supports were tested in the hydrogenation of DNT, it is found in the manuscript as: „The tests revealed that even without the noble metal the amine-functionalized cobalt ferrite, cadmium ferrite and nickel ferrite catalyst supports showed activity in TDA synthesis. The highest DNT conversion (63.01% n/n) was achieved in the case of the CoFe2O4-NH2 sample. The other two spinels were less active, 19.4 % n/n (CdFe2O4-NH2) and 26.7 % n/n (NiFe2O4-NH2) DNT conversions were measured.”
Point 4: The authors used both yields and selectivity in the figures and discussion. Please be coherent throughout the work.
Response 4: Thank you for the comment! We know that the presentation of the results is redundant and selectivity can be calculated from conversion and yield data (and vice versa). However this kind of presentation shows more information without the need of calculation of the respective data.
Reviewer 5 Report
The paper entitled “Palladium decorated, amine functionalized Ni-, Cd- and Co-ferrite nanospheres as novel and effective catalysts for 2,4-dinitrotoluene hydrogenation” prepared by Hajdu et al e developed mag- netizable, amine functionalized ferrite supported palladium catalysts. Cobalt ferrite (CoFe2O4-NH2), nickel ferrite (NiFe2O4-NH2) and cadmium ferrite (CdFe2O4-NH2) magnetic catalyst supports were produced by coprecipitation/sonochemical method. The work is interesting and the results support the claims. By the way, I recommend a major revision and it is needed to address the following issue:
-In the experimental part, more details should be given about
-please add nanostructure to the keyword
- eight references for just a paragraph is too much, please revise (line 39)
-the aim and objective are not very clear. Please revise and make it clear and concise.
-please revise the sentences and remove the grammatical errors
-Line145, please make it clear and concise
-what is the suggestion of this study for future works?
-There are many studies investigating the importance of the topic , Please add these references to your introduction and discussion parts of the manuscript and compare and bold your study novelty: https://doi.org/10.1016/j.jhazmat.2020.122296, https://doi.org/10.1021/acs.est.1c02970, DOI https://doi.org/10.1039/D1CY00467K, https://doi.org/10.1016/j.msec.2018.07.010
-The NPs stability and biocompatibility need to investigate by the authors
-It is better to compare the results of the present paper with previous works.
Author Response
Response to Reviewer 5 Comments
Comments and Suggestions for Authors
paper entitled “Palladium decorated, amine functionalized Ni-, Cd- and Co-ferrite nanospheres as novel and effective catalysts for 2,4-dinitrotoluene hydrogenation” prepared by Hajdu et al e developed mag- netizable, amine functionalized ferrite supported palladium catalysts. Cobalt ferrite (CoFe2O4-NH2), nickel ferrite (NiFe2O4-NH2) and cadmium ferrite (CdFe2O4-NH2) magnetic catalyst supports were produced by coprecipitation/sonochemical method. The work is interesting and the results support the claims. By the way, I recommend a major revision and it is needed to address the following issue:
Thank You for your valuable comments. We believe they will help to improve the quality of our manuscript!
Point 1: In the experimental part, more details should be given about
Response 1: Thank you for your suggestion. A detailed description of the hydrogenation reactor system is very important for the reproducibility of our experiments. Thus, the schematic diagram and picture of the system with main parts have been added to the manuscript as supplementary information (FigS1. and FigS2).
Point 2: please add nanostructure to the keyword
Response 2: The keyword has been added.
Point 3: eight references for just a paragraph is too much, please revise (line 39)
Response 3: The references were corrected.
Point 4: the aim and objective are not very clear. Please revise and make it clear and concise.
Response 4: Our aim was to develop catalytically highly active, stable, selective catalysts, which are easily separable from the reaction-medium, without loss. Owing to the catalytic efficacy and the magnetic features of our ferrite supported palladium catalyst, the above listed expectations have been met.
The following section was modified accordingly:
„ The aim of this work is the development of stable and selective catalysts of high activity, which are easily separable from the reaction-medium, without loss. Hereby we report the preparation and possible application of amine-functionalized magnetic, Co-, Ni-, or Cd-ferrite based catalysts decorated with Pd. The composition, morphology and surface of the nanoparticles have been examined in detail. In addition, their applicability in the industrially important hydrogenation of 2,4-dinitrotoluene has been studied.”
Point 5: please revise the sentences and remove the grammatical errors
Response 5: The grammatical errors were corrected.
Point 6: Line145, please make it clear and concise
Response 6: The mentioned sentence was clarified in the manuscript. „ For the TEM measurement, the samples were dispersed in distilled water, and this aqueous dispersion was dropped on 300 mesh copper grids (Ted Pella Inc., 4595 Redding, CA 96003, USA).”
Point 7: what is the suggestion of this study for future works?
Response 7:
Based on our experimental results and the efficiency of the developed magnetic catalysts, the conclusion was completed by the follow statement:
„ Based on our results, the above detailed magnetically separable catalysts may be well used for the hydrogenation of other aromatic nitro compounds.”
Point 8: There are many studies investigating the importance of the topic , Please add these references to your introduction and discussion parts of the manuscript and compare and bold your study novelty: https://doi.org/10.1016/j.jhazmat.2020.122296, https://doi.org/10.1021/acs.est.1c02970, DOI https://doi.org/10.1039/D1CY00467K, https://doi.org/10.1016/j.msec.2018.07.010
Response 8: Thank you for the suggestion, we modified the introduction.
˝Weng and co-workers [19]converted nickel-laden electroplating slurry into NiFe2O4 nanomaterial using sodium carbonate by a hydrothermal washing strategy. The prepared nanoparticles showed stable electrochemical Li storage performance. This new strategy can provide a sustainable approach for the conversion of heavy metals in industrial waste into high-value functional materials and for the selective recycling of heavy metals. Ebrahimi and co-workers [20]prepared superparamagnetic CoFe2O4NPs@Mn-Organic Framework core-shell nanocomposites by a layer-by-layer method. The structures exhibit high temperature stability and good magnetization. This magnetic nanometal-organic framework is an excellent candidate in targeted drug-delivery systems.˝
Point 9: The NPs stability and biocompatibility need to investigate by the authors
Response 9: The stability of the produced catalysts was tested during the reuse tests. The catalytic activity of the two tested catalyst showed no decrease. Moreover, the palladium content was measured by ICP method, and we not experienced significant palladium leaching after the reuse tests (Table 3.). In this sense, the catalyst NPs are stable, strong interaction formed between the palladium and ferrite particles. The biocompatibility of these nanoparticles were not examined, because these are made for application in heterogeneous catalysis use instead of biological application. In this respect the biocompatibility was not relevant.
Point 10: It is better to compare the results of the present paper with previous works.
Response 10:
In addition to the very high TDA selectivity and yield, the stability of the amine functionalized Pd/NiFe2O4 and Pd/CoFe2O4 catalysts is also remarkable. Compared to the amine functionalized ferrite supported catalyst, other ferrite supported catalyst (Pd/ZnFe2O4), show significant palladium leaching after the reuse tests. In the case of the non-functionalized Pd/ZnFe2O4 catalyst palladium loss after the 4th cycle was observed, the initial palladium content decreased from 4.20 wt% to 1.8 wt% [https://doi.org/10.1016/j.jmrt.2022.06.113]. This palladium loss led to a decrease in the catalytic activity during the reuse tests.
Moreover, the following section was added in the manuscript right before the Conclusions section for comparison of the experimental results.
“The activity, yield and selectivity of our catalysts are as high as of those found in the literature. Malyala and co-workers [37] investigated a powdered Y zeolite (10 % Ni/HY) catalyst containing 10 % Ni for the hydrogenation of 2,4 DNT with 85 % TDA yield and selectivity. Ren et al [38] investigated several Pd/C and Pt/ZrO2 catalysts. The Pd/C catalyst’s yield and selectivity were 98% and 99.2%, respectively. The Pt/ZrO2 cat-alysts yield varied between 97.1% and 98.9% depending upon Pt-content and reduction temperature. The selectivity values are the same as yields since conversion was 100% in each case. In addition, our catalysts are magnetic, therefore they can be easily separated from the reaction medium and even the support itself shows catalytic activity.”
Round 2
Reviewer 5 Report
The paper can be accepted